# Snf1/AMPK promotes the formation of Kog1/Raptor-bodies to increase the activation threshold of TORC1 in budding yeast

James E Hughes Hallett, Xiangxia Luo, Andrew P Capaldi*

Department of Molecular and Cellular Biology, University of Arizona, Tucson, United States

**Abstract** The target of rapamycin complex I (TORC1) regulates cell growth and metabolism in eukaryotes. Previous studies have shown that nitrogen and amino acid signals activate TORC1 via the small GTPases, Gtr1/2. However, little is known about the way that other nutrient signals are transmitted to TORC1. Here we report that glucose starvation triggers disassembly of TORC1, and movement of the key TORC1 component Kog1/Raptor to a single body near the edge of the vacuole. These events are driven by Snf1/AMPK-dependent phosphorylation of Kog1 at Ser 491/494 and two nearby prion-like motifs. Kog1-bodies then serve to increase the threshold for TORC1 activation in cells that have been starved for a significant period of time. Together, our data show that Kog1-bodies create hysteresis (memory) in the TORC1 pathway and help ensure that cells remain committed to a quiescent state under suboptimal conditions. We suggest that other protein bodies formed in starvation conditions have a similar function.

*For correspondence: capaldi@email.arizona.edu

**Competing interests:** The authors declare that no competing interests exist.

## Introduction

The Target Of Rapamycin Complex I (TORC1) is made up of three essential proteins, called mTOR, Raptor and mLST8 in humans and Tor1, Kog1 and Lst8 in yeast (*Kim et al., 2002*; *Loewith et al., 2002*). Biochemical and structural studies show that these proteins form a stable ring structure, containing two copies of each subunit (*Adami et al., 2007*; *Yip et al., 2010*). Kog1/Raptor is known to recruit substrates to the TOR kinase (Tor1) and is required for regulation of Tor1 activity (*Hara et al., 2002*; *Kim et al., 2002*). Lst8, on the other hand, binds directly to the kinase domain in Tor1 (*Yip et al., 2010*) and may help stabilize the interaction between Kog1 and Tor1 (*Kim et al., 2002*). In yeast, TORC1 also includes the non-essential (and poorly characterized) protein Tco89 (*Reinke et al., 2004*).

In the presence of pro-growth hormones and the appropriate nutrients, TORC1 is active and drives protein, lipid, and nucleotide synthesis by phosphorylating a wide range of proteins (*Bodenmiller et al., 2010*; *Hsu et al., 2011*; *Laplante and Sabatini, 2012*; *Loewith and Hall, 2011*; *Robitaille et al., 2013*). In contrast, when hormone, nutrient or energy levels drop below a critical level, TORC1 is inhibited, causing the cell to switch from anabolic to catabolic metabolism and eventually enter a quiescent state (*Barbet et al., 1996*; *Duvel et al., 2010*; *Loewith and Hall, 2011*).

In higher eukaryotes, hormone signals are transmitted to TORC1 through the Phosphatidylinositol 3-kinase to AKT, which in turn phosphorylates the tuberous sclerosis complex (*Inoki et al., 2002*; *Manning et al., 2002*). The tuberous sclerosis complex (TSC1, TSC2 and TBC1D7) then dissociates from the lysosomal surface, triggering activation of the small GTPase, Rheb, and subsequently TORC1 (*Dibble and Manning, 2013*; *Menon et al., 2014*). However, these pro-growth signals are

**eLife digest** In humans, yeast and other eukaryotes, a group of proteins called the Target of Rapamycin Complex I (TORC1) promote cell growth and increase metabolic activity when nutrients are plentiful. Previous studies have shown how molecules that contain the nutrient nitrogen – which is needed to make proteins – activate TORC1. However, it is not clear how other nutrients regulate this complex.

Bakers yeast is a simple, single celled organism that researchers often use as a model to study how cells work. The yeast TORC1 is made up of three core proteins, including Kog1 and Tor1. Kog1 selectively recruits proteins to the complex, where they are modified by Tor1 to regulate their activity. Here, Hughes Hallett et al. used microscopy to study what effect sugar starvation has on the complex.

In the experiments, yeast cells were genetically engineered so that Kog1 and Tor1 appeared fluorescent under the microscope. The experiments reveal that, when sugar is in short supply, Kog1 breaks away from the rest of the TORC1 and moves to another part of the cell where it accumulates to form a cluster called a "body". This movement is driven by a "kinase" enzyme that adds chemical groups called phosphates to Kog1, and by regions within the Kog1 protein known as prion like domains. When sugar first becomes available again, Kog1 is still in the body so Tor1 cannot immediately trigger cell growth. However, once a steady supply of sugar resumes, Kog1 rejoins the rest of the complex and the cells start to grow.

Together, Hughes Hallett et al.'s findings reveal that the formation of Kog1 bodies during sugar starvation creates a "memory" that prevents TORC1 from reactivating cell growth if sugar is only temporarily available. Humans have over 100 proteins that contain prion like domains. Therefore a future challenge is to find out whether any of these proteins form similar bodies that enable our cells to remember past events.

blocked in energy starvation conditions due to stimulation of the AMP activated protein kinase (AMPK), which phosphorylates and hyperactivates TSC1-TSC2 to inhibit Rheb and TORC1 (*Inoki et al., 2003a*; *Inoki et al., 2003b*).

AMPK also inhibits TORC1 by phosphorylating Kog1/Raptor at two sites (*Gwinn et al., 2008*). One of these sites is highly conserved in eukaryotes (Ser 959 in *S. cerevisiae*) but does not appear to influence TORC1 activity in budding yeast (*Kawai et al., 2011*).

In both higher and lower eukaryotes, amino acid/nitrogen signals are transmitted to TORC1 through another class of small GTPases, known as Gtr1/2 in yeast and the Rags in humans (*Binda et al., 2009*; *Kim et al., 2008*; *Sancak et al., 2008*). Gtr1/2 are tethered to the vacuolar/lysosomal membrane via the EGO complex (Ragulator in humans) and the vacuolar ATPase, where they respond to amino acid signals by binding and activating TORC1 (*Bar-Peled et al., 2012*; *Binda et al., 2009*; *Panchaud et al., 2013*; *Sancak et al., 2010*; *Zoncu et al., 2011*).

Outside of nitrogen/amino acid starvation conditions, little is known about the way nutrient and stress signals trigger changes in TORC1 activity, particularly in organisms such as *S. cerevisiae* that are missing TSC1/2 and where there is no clear link between AMPK and TORC1 activity. For example, studies in yeast have shown that glucose starvation completely blocks TORC1 signaling, but only around 20% of this inhibition depends on Gtr1/2 and the other known TORC1 regulator in yeast, Rho1 (*Hughes Hallett et al., 2014*).

Here, to learn more about TORC1 regulation, we examine the localization of Tor1 and Kog1 in budding yeast, in glucose starvation and other conditions. Surprisingly, we find that glucose starvation leads to disassembly of the TOR complex and movement of Kog1 from the vacuolar membrane to a single body on the edge of the vacuole, while glucose repletion leads to a reversal of this process. Following up on these observations we show that the AMPK, Snf1, drives Kog1 into bodies by phosphorylating, or triggering the phosphorylation of, Kog1 at Ser 491 and 494. Furthermore, we find that these phosphorylation sites are located in one of two, glutamine-rich prion-like motifs in Kog1, and that these motifs are required for the formation of Kog1-bodies. Finally, by measuring phosphorylation of the key TORC1 substrate, Sch9, in a variety of mutants, we show that Kog1 agglomeration increases the threshold for TORC1 activation.

Taken together, our results reveal a novel mechanism of TORC1 regulation and show that protein body formation can block the activation of a pathway by weak stimuli. The numerous other protein bodies formed in yeast, and other organisms, may have a similar function—creating hysteresis (memory) in key signaling and metabolic pathways to ensure that once cells commit to a stress or starvation state, they only exit when conditions are optimal.

## Results

### Movement of Kog1 into bodies

To determine if TORC1 localization is altered in glucose starvation conditions, we subjected yeast carrying Kog1-YFP, Tco89-YFP (see Material and methods), or Tor1 with a triple GFP insertion (*Binda et al., 2009*; *Sturgill et al., 2008*) to acute glucose starvation and acquired 3D images using a fluorescence microscope. Surprisingly, we found that Kog1 and Tco89 move from their known location on the vacuolar membrane (marked with Vph1-mCherry), to a single body near the edge of the vacuole, within 60 min in most cells (*Figure 1A* and *Figure 1—figure supplement 1*). In contrast, Tor1 remained associated with the vacuolar membrane and/or moved to the cytoplasm during the same time-period (*Figure 1B*).

To study Kog1-body formation in more detail, we next followed Kog1-YFP localization in cells transferred from rich medium (SD) to glucose starvation, nitrogen starvation, glucose + nitrogen starvation, and osmotic stress conditions. In glucose starvation and nitrogen + glucose starvation we saw rapid formation of Kog1-bodies ( $\tau = 11.0 \pm 2.8$ and $13.6 \pm 2.4$ min) in $57 \pm 5\%$ and $73 \pm 4\%$ of cells, respectively (*Figure 1C*). Kog1-bodies also formed in nitrogen starvation conditions, albeit at a much slower rate ($\tau = 153 \pm 25$ min; *Figure 1C*). However, we did not see a significant increase in Kog1-body formation in osmotic stress or when cells were simply transferred into fresh SD medium (*Figure 1C*).

In separate experiments, we also followed Kog1-body formation during long periods of glucose and/or nitrogen starvation (2-–6 hrs). In all cases, we saw slow accumulation of Kog1-bodies, beyond the levels found at one hour, until 80-85% of cells had a least one, but on occasion 2 or 3, Kog1 bodies (*Figure 1—figure supplement 2, 3*). Cultures grown to saturation in SD medium also formed Kog1-bodies in ~~85% of cells (*Figure 1—figure supplement 4*).

Finally, to determine if Kog1-body formation is a reversible process, we exposed cells to glucose starvation conditions for 60 min and then followed Kog1-YFP localization after adding 0.02%, 0.1%, or 2% glucose back to the medium. In both 0.1% and 2% glucose, we saw Kog1 move out of the bodies and back onto the vacuolar membrane over an approximately one-hour time-period ($\tau = 48 \pm 8$ and $45 \pm 10$ min, respectively; *Figure 1D*). This relocalization also occurred in the presence of cycloheximide, demonstrating that existing Kog1-bodies dissociate during glucose repletion (*Figure 1D* and *Figure 1—figure supplement 5*).

Putting these data together we conclude that Kog1-body formation is a rapid ( $\tau = \sim10$ min) and reversible process, triggered by glucose and nitrogen starvation.

### Kog1-bodies are distinct from stress granules and P-bodies

Previous work has shown that TORC1 can localize to stress granules in both yeast and human cells (*Takahara and Maeda, 2012*; *Thedieck et al., 2013*; *Wippich et al., 2013*). To determine if Kog1 associates with stress granules (or related structures known as P-bodies [*Parker, 2012*]) in glucose starvation conditions, we transfected cells carrying Kog1-YFP with mCherry-tagged versions of the stress granule marker Pbp1 or the P-body marker Edc3, and followed YFP and RFP localization. We found that stress granules start to form after 60 min in glucose starvation conditions, and become pervasive in cells grown to saturation in SD medium, but only co-localize with Kog1-bodies in 2.6% of cells (0% co-localization after 60 min, n = 94; 2.6% co-localization after 24 hrs, n = 190; *Figure 2A*). Similarly, P-bodies form in >80% cells after 60 min of glucose starvation, but only co-localize with Kog1-bodies in 3.4% of cells (n = 404; *Figure 2B*). Therefore, it appears that the Kog1-bodies formed in starvation conditions are distinct from both stress granules and P-bodies.

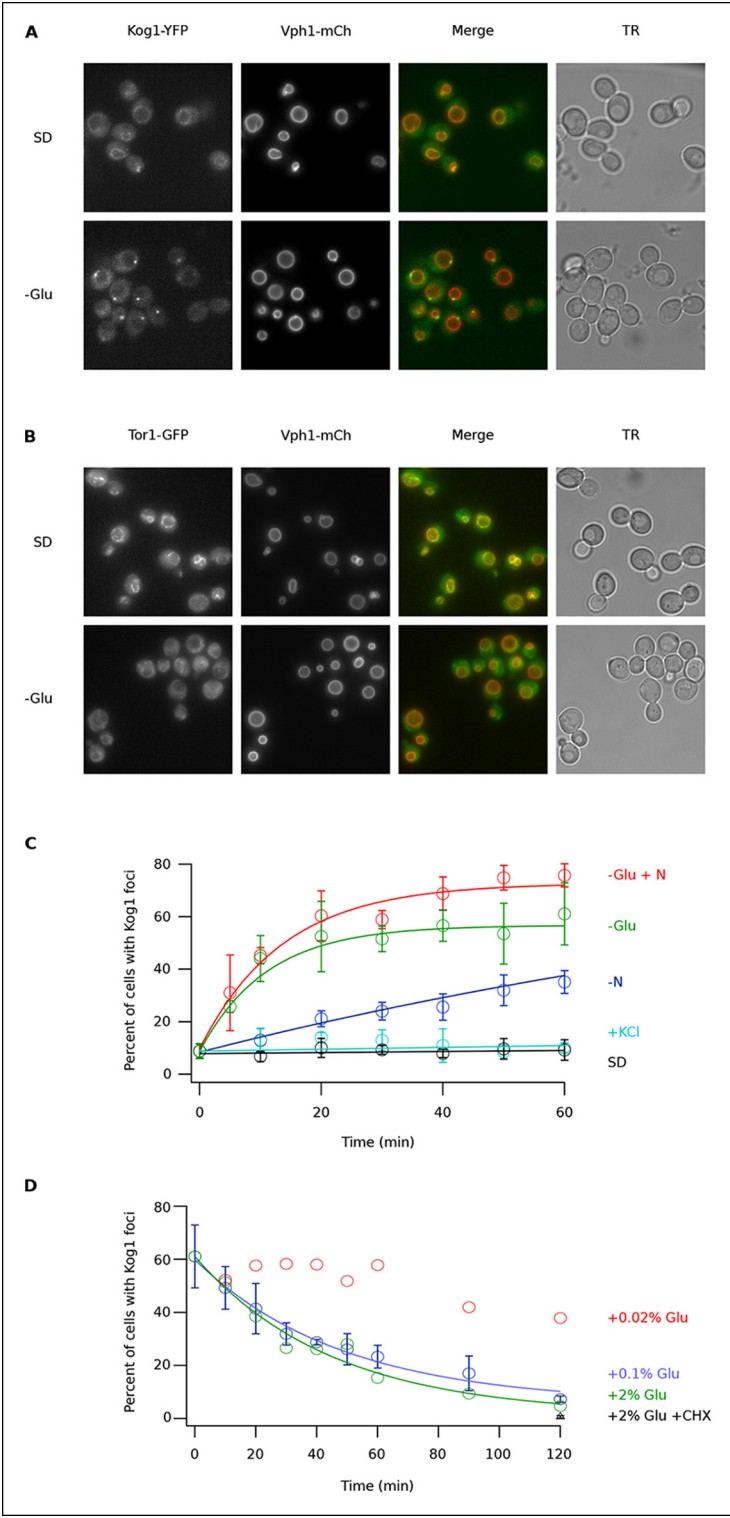

**Figure 1.** Kog1-YFP moves from the vacuolar membrane to a single body during glucose and nitrogen starvation. (**A**) Localization of Kog1-YFP and Vph1-mCherry, before (SD) and after (-Glu) glucose withdrawal (60 min). (**B**) Localization of Tor1-3xGFP and Vph1-mCherry before (SD) and after (-Glu) glucose withdrawal (60 min). (**C**) Time-course data showing the fraction of cells that contain Kog1-bodies after transfer to synthetic medium with 2% glucose (SD), SD + 0.4 M KCl (+ KCl), SD -glucose (-Glu), SD-glucose and nitrogen (-Glu + N), and SD-nitrogen (-N) medium. Each time-point shows the average and standard deviation from three independent experiments (performed on different days with >200 cells per time-point, per replicate). Solid lines show the best fit to a single

*Figure 1. continued on next page*

*Figure 1. Continued*

exponential equation (-Glu, -N, and –Glu + N) or a line (SD and + KCl). (**D**) Time-course data showing the fraction of cells that contain Kog1-bodies after adding glucose, or glucose and cycloheximide (0.02% Glu, 0.1% Glu, 2% Glu or 2% Glu + 100 ug/ml cycloheximide) back to cells that have been in SD–glucose for 60 min. Each datapoint in 0.1% glucose and 2% + 100 ug/ml cycloheximide shows the average and standard deviation from three independent experiments (performed on different days with >200 cells per time-point, per replicate). The solid lines show the best fit to a single exponential equation.

The following figure supplements are available for Figure 1:

**Figure supplement 1.** The TORC1 component Tco89 associates with Kog1-bodies.

**Figure supplement 2.** Kog1-body formation is pervasive during extended periods of nutrient starvation.

**Figure supplement 3.** Kog1 forms one body per cell.

**Figure supplement 4.** Kog1-body formation is pervasive following exit of exponential growth.

**Figure supplement 5.** Kog1-YFP dissociates from perivacuolar bodies and relocalizes to the vacuolar membrane upon glucose repletion.

## Snf1 regulates Kog1-body formation

Once we determined that Kog1-body formation occurs independently of stress granule and P-body formation, we sought to identify the signal that triggers Kog1 agglomeration in glucose starvation conditions. In a previous study we showed that the AMP activated protein kinase, Snf1, is required for TORC1 pathway inhibition during glucose starvation (*Hughes Hallett et al., 2014*). We therefore wondered if Snf1 regulates TORC1, at least in part, by driving Kog1 agglomeration. To test this hypothesis we monitored Kog1-YFP localization in *snf1Δ* cells. These experiments revealed that deletion of Snf1 causes a dramatic, 20-fold, increase in the time-constant for Kog1-body formation ( $\tau$ = 219 ± 21 min in *snf1Δ* cells; *Figure 3A*).

Next, to determine how Snf1 activates Kog1-body formation, we purified TORC1 from cells: (1) growing in SD medium, (2) exposed to glucose starvation conditions for 5 min, and (3) exposed to osmotic stress conditions for 5 min (see Material and methods). We then used mass spectrometry to identify the phosphorylation sites on Tor1 and Kog1 in each condition. These data showed that Kog1 is phosphorylated on Ser 491 and 494 in glucose starvation conditions, but not in osmotic stress or SD medium (*Figure 3—figure supplements 1, 2*). A short time later, Young and coworkers showed that Ser 491 and 494 on Kog1 are Snf1-dependent phosphorylation sites (*Braun et al., 2014*). Therefore, to test if Snf1 drives Kog1-body formation by promoting phosphorylation of Ser 491 and 494, we constructed a strain carrying Kog1$^{S491A/S494A}$-YFP at the native locus and monitored Kog1 localization in glucose starvation conditions. We found that Kog1$^{S491A/S494A}$ forms bodies slowly ( $\tau\tau$ = 309 ± 48 min; *Figure 3B*), and at a rate similar to that found in the *snf1Δ* strain ($\tau$ = 219 ± 21 min; *Figure 3A*). Thus, Snf1 increases the rate of Kog1 body formation by phosphorylating, or triggering phosphorylation, of Kog1 in glucose starvation conditions.

Kog1$^{S491A/S494A}$ cells also showed a reduced level of Kog1-body formation in SD medium (2.3 ± 0.1% bodies in Kog1$^{S491A/S494A}$ cells versus 8.6 ± 2.8% in wild type cells; *Figure 3B*). This was not seen in the *snf1Δ* strain (8.3 ± 0.8% bodies in SD; *Figure 3A*), suggesting that either an additional kinase phosphorylates Kog1 at positions 491 and 494 in log growth conditions or that mutation of Ser 491 and 494 to alanine also inhibits Kog1 agglomeration by altering the physical properties of Kog1.

Finally, to test the impact that constitutive phosphorylation of Kog1 has on Kog1-body formation; we attempted to create strains carrying the phosphomimetic variants Kog1$^{S491D/S494D}$ and Kog1$^{S491E/S494E}$. However, these mutants were not viable, indicating that Kog1-body formation (or Kog1 phosphorylation) inhibits cell growth and/or division (*Figure 3—figure supplement 3*).

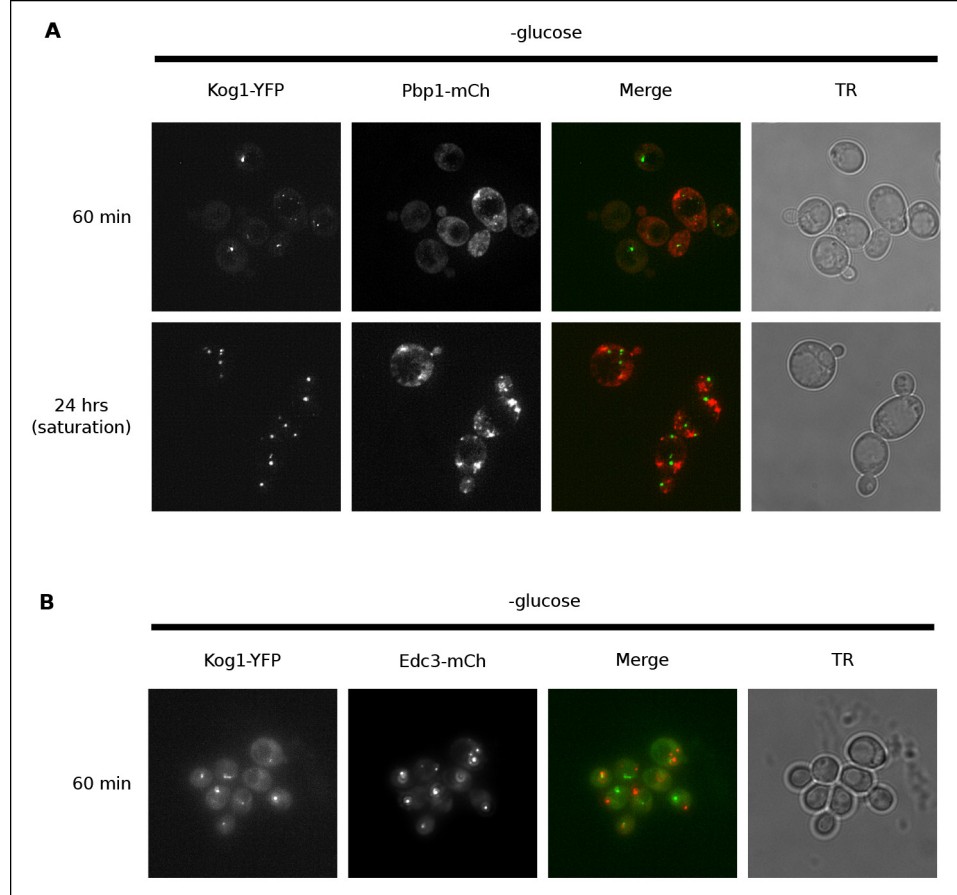

**Figure 2.** Kog1-YFP does not associate with stress granules or P-bodies in glucose starvation conditions. (**A**) Localization of Kog1-YFP after 60 min or 24 hrs of glucose starvation in cells expressing the stress granule marker Pbp1-mCherry. (**B**) Localization of Kog1-YFP after 60 min of glucose starvation in cells expressing the P-Body marker Edc3-mCherry. The images in (**A**) were deconvolved using Deltavision software to ensure we could distinguish Pbp1 granules from cytoplasmic Pbp1.

## Kog1-body formation depends on prion-like domains in Kog1

Protein body formation can be driven by called prion-like motifs or PriLMs (*Alberti et al., 2009*; *Decker et al., 2007*; *Gilks et al., 2004*; *Han et al., 2012*). These motifs tend to have long stretches of glutamine and/or asparagine residues and a low number of hydrophobic and/or charged residues (*Alberti et al., 2009*). In examining the Kog1 sequence we identified two such motifs, both containing long stretches of glutamine (*Figure 4A*). These regions were also identified as PriLMs using a hidden Markov Model trained on known prion sequences (*Alberti et al., 2009*). Interestingly, the first of these PriLMs includes the Snf1 dependent phosphorylation sites at Ser 491 and 494, while the second, smaller, PriLM is located approximately 300 amino acids away (*Figure 4A*).

To determine if the PriLMs in Kog1 are involved in Kog1-body formation we mutated a stretch of glutamines in each motif, to form a stretch of alanines (*Figure 4A*), and measured the impact on Kog1-YFP localization. Disruption of prion-like motif 1 (PrDm1) caused a >2-fold decrease Kog1-body formation, both in SD medium ($2.76 \pm 0.6\%$ in PrDm1 versus $8.6 \pm 2.8\%$ in wt; *Figure 4B*) and in glucose starvation conditions ($26 \pm 2.7\%$ in PrDm1 versus $57 \pm 5\%$ in wt; *Figure 4B*). Disruption of prion-like motif 2 (PrDm2) also caused a (small) decrease Kog1-body formation in glucose starvation conditions ($45 \pm 4\%$ in PrDm2 versus $57 \pm 5\%$ bodies in wt; *Figure 4B*). More remarkably, however, disruption of PriLM1 and PriLM2 completely blocked Kog1-body formation in both SD medium and glucose starvation conditions ($1.0 \pm 2.1\%$ cells with bodies; *Figure 4B*). Thus, Kog1-body formation depends on the two prion-like motifs in Kog1.

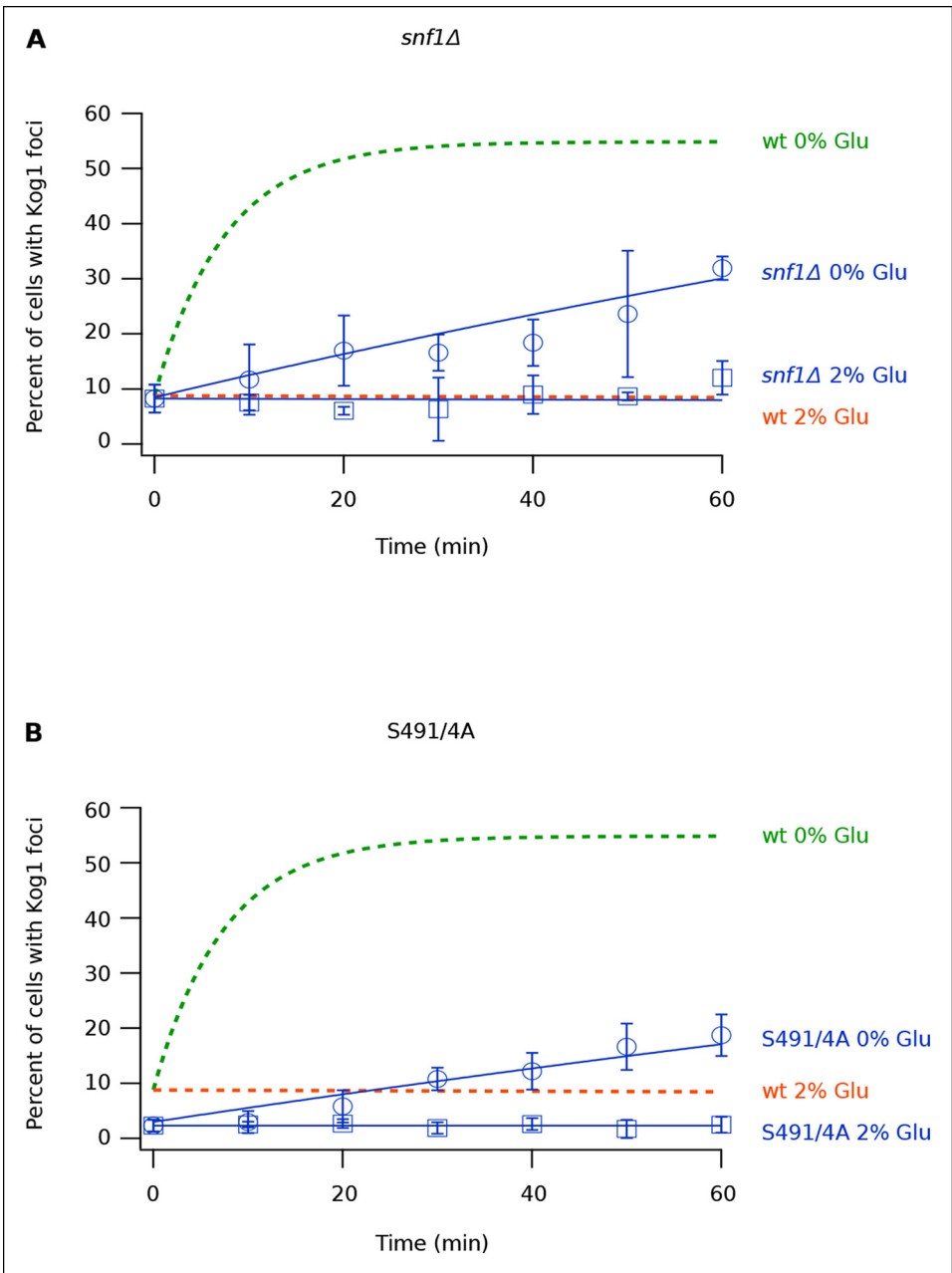

**Figure 3.** Snf1 Dependent Phosphorylation of Kog1 drives Kog1-body formation. (**A——B**) Time-course data showing the fraction of (**A**) *snf1Δ* and (**B**) Kog1$^{S491A/S494A}$ cells that contain Kog1-bodies in SD medium (2% Glu) and at various time-points after glucose withdrawal (0% Glu). The blue points show the average and standard deviation from three independent experiments (performed on different days with >200 cells per time-point, per replicate), while the blue lines show the best fit to a single exponential equation (0% Glu) or a line (2% Glu). The green and orange lines show the best fit to the wild-type data (from *Figure 1C*) for comparison.

The following figure supplements are available for Figure 3:

**Figure supplement 1.** Mass spectrometry generated peptide map for Kog1.

**Figure supplement 2.** Protein identification result identifying S491 and S494 as sites of phosphorylation.

**Figure supplement 3.** Strains carrying Kog1$^{S491D/S494D}$ and Kog1$^{S491E/S494E}$ at the native loci are inviable.

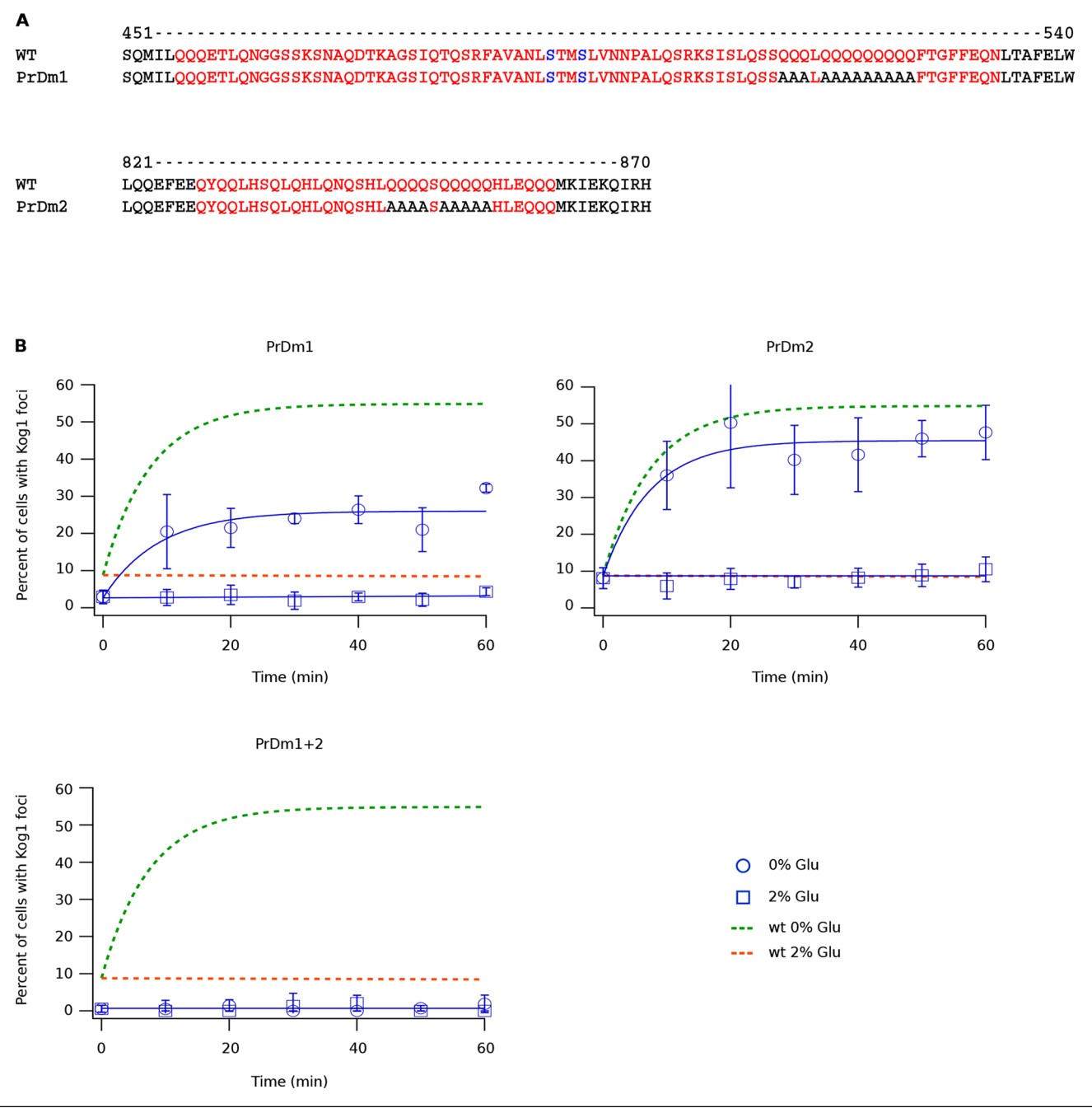

**Figure 4.** Kog1-body Formation Depends on Prion-Like Domains in Kog1. (**A**) Sequence of two prion-like motifs in Kog1 (red letters) as defined by Alberti and coworkers (*Alberti et al., 2009*). The Snf1 dependent phosphorylation sites at Ser 491 and Ser 494 are shown in blue. (**B**) Time-course data showing the fraction of cells carrying Prion Domain mutation 1 (PrDm1), Prion Domain mutation 2 (PrDm2), or both mutations (PrDm1 + 2), that have Kog1-bodies in SD medium (0% Glu), and after glucose withdrawal (2% Glu). The blue points show the average and standard deviation from three independent experiments (performed on different days with >200 cells per time-point, per replicate) while the blue lines show the best fit of each dataset to a single exponential equation (0% Glu for PrDm1 and PrDm2) or a line. The green and orange lines show the best fit to the wild-type data (from *Figure 1C*) for comparison.

## Kog1-body formation increases the threshold for TORC1 activation

During log growth, TORC1 phosphorylates and activates the S6 kinase, Sch9, to drive protein and ribosome synthesis (*Bodenmiller et al., 2010*; *Huber et al., 2011*; *Urban et al., 2007*). This key

signaling event is readily measured using a band-shift assay developed by Loewith and coworkers (*Urban et al., 2007*). Using this assay, we asked if and how Kog1-bodies influence TORC1 signaling by following Sch9 phosphorylation in five strains that have a defect in Kog1-body formation (Kog1$^{S491A/S494A}$, *snf1Δ*, PrDm1, PrDm2, PrDm1 + 2).

We found that Sch9 is phosphorylated at or near wild-type levels during log phase growth in all of our mutant strains (*Figures 5A, B*). Sch9 is then rapidly (≤2.5 min) and completely dephosphory-lated during glucose starvation in the wild-type, Kog1$^{S491A/S494A}$, PrDm1, PrDm2 and PrDm1 + 2 strains (*Figures 5A, B*). Thus, Kog1-body formation does not appear to play a significant role in TORC1-Sch9 signaling during acute glucose starvation since (1) Sch9 dephosphorylation occurs much faster than the accumulation of Kog1-bodies ( ττ = <2.5 min versus 11 ± 3 min) and (2) most of the mutants that have a defect in Kog1-body formation behave like the wild-type strain in the Sch9 phosphorylation assay. The only exception is the *snf1Δ* strain, where we see a clear defect in Sch9 dephosphorylation after 5 and 10 min of glucose starvation (*Figure 5A*). However, it is very unlikely that this defect is due to inhibition of Kog1-body formation, since the Kog1$^{S491A/S494A}$ strain does not have a defect in the Sch9 assay (*Figure 5A*). Instead, it appears that Snf1 regulates the TORC1 pathway by triggering Kog1 phosphorylation at Ser 491 and 494, and through an additional, unknown, mechanism.

After discovering that TORC1/Sch9 inhibition precedes Kog1-body formation, we reasoned that Kog1 agglomeration might act as a slow step to lock TORC1 in an inactive state. To test this hypothesis we subjected wild-type and mutant strains to acute glucose starvation for 60 min, added 2% glucose back to the medium, and then followed Sch9 phosphorylation (*Figure 5A*). To our surprise, we saw complete, or near complete, Sch9 phosphorylation in all of our strains (*Figures 5A, B*). However, when we repeated this experiment, but only added 0.02% or 0.1% glucose back to the medium, we saw significantly more Sch9 phosphorylation in the mutant cells than in wild-type cells (*Figures 5C, D* and *Figure 5—figure supplement 1 and 2*). In fact, 0.1% glucose triggered ~50% Sch9 phosphor-ylation in the wild-type strain compared with ~100% Sch9 phosphorylation in our strongest mutant, PrDm1 + 2 (*Figure 5D*). These results are remarkable since only 60% of wild-type cells have Kog1-bodies in our starting conditions (*Figure 1C*) and as a result, it should only be possible to see a 60% increase in Sch9 phosphorylation in the mutant strains (an increase from 40% activity in wild-type cells to 100% activity in mutant cells). Therefore, putting all of our band-shift data together, we conclude that Kog1-body formation acts to increase the threshold for TORC1 activation in cells subjected to medium or long-term glucose starvation.

## Discussion

In this study we show that glucose starvation leads to rapid inactivation of TORC1/Sch9 signaling (τ<2.5 min, *Figure 5*), dissociation of the Kog1-Tor1 complex, and the formation of Kog1-bodies (τ = 11 ± 3 min). Kog1-bodies then act to increase the threshold for TORC1 activation–probably by limiting the number of intact TORC1 molecules in the cell. In other words, Kog1-body formation helps ensure that cells commit to a starvation or quiescent state, once they have been in glucose starvation conditions for a significant period of time. This is likely important for survival in natural conditions where yeast cells are exposed to a complex and fluctuating environment, and may need to remain in a quiescent state for years (*Loewith and Hall, 2011*).

We also show that Kog1-body formation is driven by the AMPK, Snf1. This kinase is inactive during log growth and then activated upon glucose starvation, where it triggers massive changes in transcription and metabolism (*Braun et al., 2014*; *Hedbacker and Carlson, 2008*; *Usaite et al., 2009*). As part of this program, Snf1 activates phosphorylation of Kog1 at Ser 491 and 494. It is important to note, however, that Snf1-dependent phosphorylation of Kog1 is probably indirect since the sequence around Ser 491 and 494 (AVANLS*TMS*LVN) does not match known AMPK targets sites such as LRRVxS*xxNL and MKKSxS*xxDV (*Gwinn et al., 2008*). Further work is therefore needed to identify the kinase that acts downstream of Snf1 to phosphorylate Kog1.

Snf1-dependent phosphorylation of Kog1 occurs in the middle of a glutamine rich, prion-like motif. We show that this prion-like motif and another similar motif, located 300 amino acids away, are essential for Kog1-body formation. Interestingly, the Snf1-dependent phosphorylation sites, and both prion-like domains found in S. *cerevisae*, are conserved in numerous other yeast species including the pathogen *Candida glabrata* (*Figure 6*). However, we could not find prion-like domains in

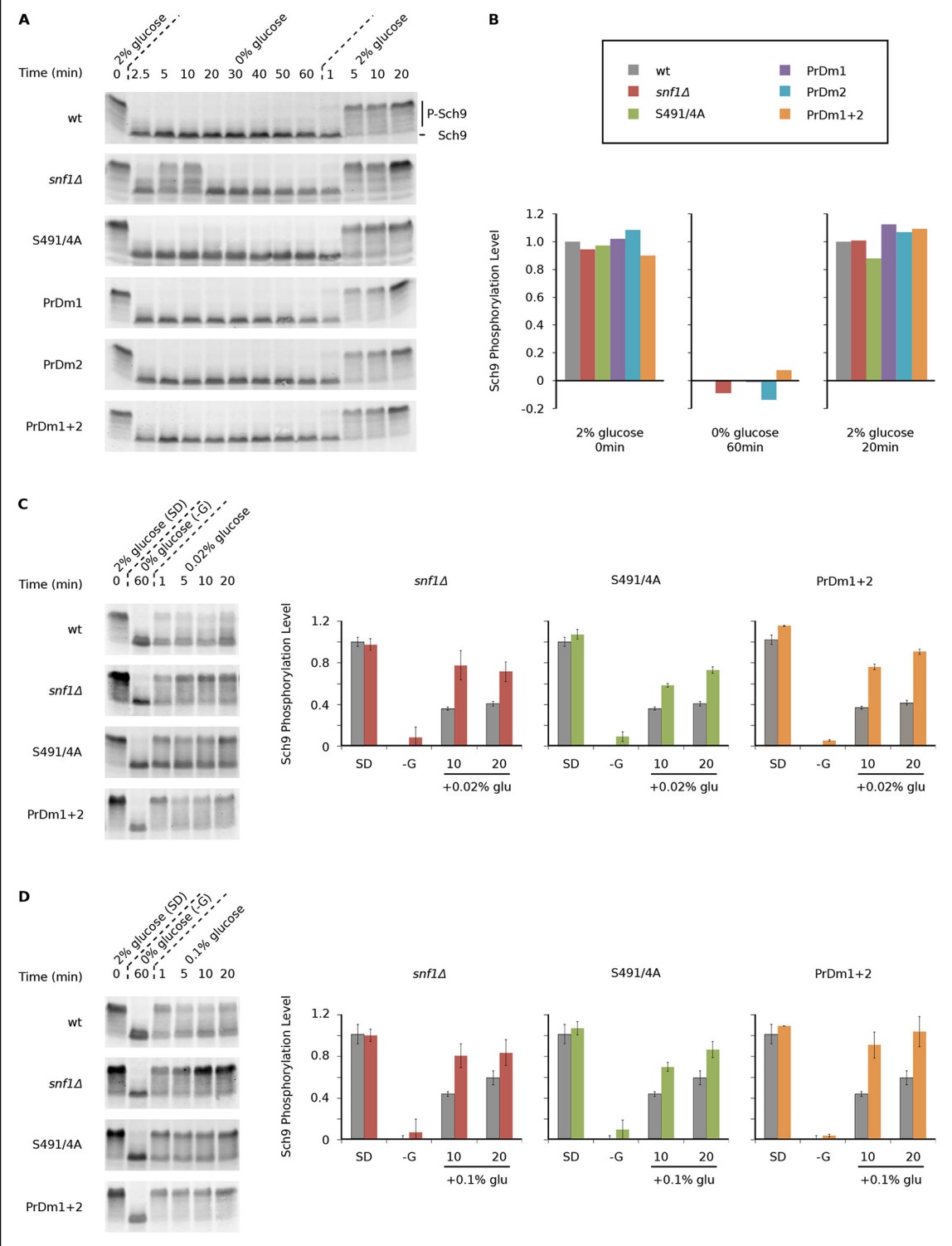

**Figure 5.** Impact of Kog1-bodies on TORC1 Signaling. (**A**) Bandshift assays measuring Sch9 phosphorylation in SD medium (0 min; 2% glucose), at various time-points after glucose withdrawal (2.5-–60min; 0% glucose), and then at various time-points after adding 2% glucose back to the cultures (1, 5, 10, and 20 min; 2% glucose). (**B**) Quantitation of the band-shift data from (**A**). The data are normalized to set the wild-type value to 1.0 in 2% glucose, and 0.0 in 0% glucose (see Methods). (**C-–D**) Sch9 phosphorylation data for wild-type, *snf1Δ*, Kog1[S491A/S494A], and PrDm1 + 2 cells, grown in SD medium (2% glucose), after transfer into synthetic medium - glucose for 60 min (0% glucose), and then at various time-points after adding 0.02% (**C**) or 0.1% glucose (**D**) back to the culture. Quantitation in (**C**) and (**D**) was performed as described in (**B**) but here the graphs show the average and standard deviation from three separate experiments. Data for PrDm1 and PrDm2 are shown in *Figure 5—figure supplement 1*.

*Figure 5. continued on next page*

*Figure 5. Continued*

The following figure supplements are available for Figure 5:

**Figure supplement 1.** Impact of Kog1-bodies on TORC1 signaling.
**Figure supplement 2.** Sch9 reactivation depends upon TORC1 activity.

Kog1 from *Schizosaccharomyces pombe, Neurospora Crassa*, and related species (*Figure 6*). Strikingly, most (5/6) of the yeast species missing prion-like domains in Kog1 have genes encoding the tuberous sclerosis complex (TSC1 and TSC2), while all (17/17) of the yeast species with prion-like domains in Kog1 are missing the tuberous sclerosis complex (TSC1 or TSC1 and TSC2 genes; *Table 1*).

We also find prion-like domains in Kog1 from *C. elegans* and other related species, but not in flies, mice or humans (*Figure 6—figure supplement 1*). Again, the organisms with prion-like domains in Kog1 are missing TSC1 and TSC2 (*Figure 6—figure supplement 1*).

The observation that most organisms either have prion-like domains in Kog1, or have genes encoding the tuberous sclerosis complex, is especially interesting when you consider the similarities between TSC1-TSC2 and Kog1-body function. Studies in human cells have shown that the TSC1-TSC2 complex serves to inhibit TORC1 in the absence of pro-growth hormones and/or the presence of AMPK activity (*Huang and Manning, 2008*; *Inoki et al., 2002*; *Inoki et al., 2003b*), while we show that Kog1-body formation serves to inhibit TORC1 in the absence of glucose and the presence of AMPK activity. It therefore appears that the mechanisms underlying TORC1 regulation diverged in early eukaryotic evolution, so that most simple eukaryotes use Kog1-body formation to regulate TORC1 signaling, while higher eukaryotes (like flies, mice and humans) use the TSC1-2 complex. Going forward, it will be important to learn more about the differences between these two regulatory mechanisms, especially as drugs targeting Kog1-bodies may be able to block the growth of fungi and worms without affecting TORC1 signaling in humans and other higher eukaryotes.

Beyond establishing a role for Kog1-body formation in TORC1 regulation, the data presented here shed light on the role that protein body formation plays in a cell. Studies in yeast, and other organisms, have shown that hundreds of proteins move into bodies, when cells are exposed to stress or starvation conditions (*Narayanaswamy et al., 2009*; *O'Connell et al., 2014*; *O'Connell et al., 2012*; *Shah et al., 2014*). For example, more than 30 proteins involved in mRNA decay and translation move into stress granules and P-bodies during long-term starvation (*Decker and Parker, 2012*). However, to date, it has been difficult to pinpoint the function of RNA granules and other bodies (*Decker and Parker, 2012*; *O'Connell et al., 2012*). We show that the movement of Kog1 into bodies acts to increase the threshold for TORC1 activation. This creates hysteresis in the TORC1 pathway and likely helps ensure that cells exposed to low levels of glucose do not try to re-enter a rapid growth state. We suggest that other protein bodies formed during stress and starvation conditions function in a similar way; setting new activation thresholds for key signaling and metabolic pathways. Further work will be needed to test this idea, and to determine if there are differences between the hysteretic behavior created by reversible protein agglomeration and other mechanisms such as feedback loops (*Ferrell, 2002*).

## Materials and methods

### Experimental procedures

#### *S. cerevisiae* strains

All strains used in this study were generated in a haploid *S. cerevisiae* strain, W303 background (*trp1, can1, leu2, his3, ura3*), using standard methods and are listed in *Table 2.* Note that the Kog1-YFP and Tor1-3xGFP strains have the same growth rate as the wild-type strain and thus Kog1-YFP and Tor1 with an internal 3xGFP are fully functional (*Binda et al., 2009*; *Sturgill et al., 2008*).

**Table 1.** PriLM1 and PriLM2 are anti-correlated with the presence of Tsc1 and Tsc2 genes in fungal genomes. Quantitation, **number** and (percentage), of Glutamine (Q) and Asparagine (N) residues within PriLM1 and PriLM2 domains of Kog1 and otologs (**Figure 6**) reveals three clusters of hosts. (1) Species carrying Q/N rich PriLM1 and PriLM2 domains lack both Tsc1 and Tsc2 genes and the tuberous sclerosis complex (TSC) signaling pathway but show conservation of S491/494 phosphorylation motif. (2) Species carrying PriLM1 of intermediate Q/N richness lack Tsc1 but carry the Tsc2 gene. The influence on TSC pathway signaling is unclear. (3) Species lacking PriLM1 and PriLM2 carry both Tsc1 and Tsc2 genes and an operational TSC signaling pathway. Genbank references to Tsc1 and Tsc2 orthologs are provided.

| | PriLM1 | | PriLM2 | | | | |
| | Q/N | (%QN) | Q/N | (%QN) | Tsc1 | Tsc2 | S491/494 |
|---|---|---|---|---|---|---|---|
| *S. cerevisiae* | 28 | (35.90) | 20 | (58.82) | | | + |
| *S. kluyveri* | 49 | (55.68) | 17 | (65.38) | | | + |
| *S. paradoxus* | 29 | (36.71) | 20 | (58.82) | | | + |
| *S. mikatae* | 28 | (36.36) | 20 | (58.82) | | | + |
| *S. castellii* | 32 | (37.21) | 15 | (55.56) | | | + |
| *S. bayanus* | 24 | (32.43) | 21 | (60.00) | | | + |
| *C. glabrata* | 21 | (26.92) | 17 | (50.00) | | | + |
| *A. gossypii* | 19 | (26.39) | 18 | (56.25) | | | + |
| *K. waltii* | 24 | (34.29) | 11 | (52.38) | | | + |
| *L. elongosporus* | 31 | (40.26) | 1 | (16.67) | | | |
| *K. lactis* | 17 | (27.42) | 24 | (63.16) | | CAH03101.1 | + |
| *C. parapsilosis* | 17 | (27.42) | 0 | (0.00) | | CCE43871.1 | |
| *C. lusitaniae* | 1 5 | (28.30) | 0 | (0.00) | | EEQ40372 | |
| *D. hansenii* | 14 | (23.73) | 1 | (50.00) | | CAG86052.2 | |
| *C. albicans* | 14 | (28.00) | 0 | (0.00) | | EAK92235.1 | |
| *C. guilliermondii* | 11 | (24.44) | 0 | (0.00) | | EDK41275.2 | |
| *C. tropicalis* | 11 | (16.42) | 0 | (0.00) | | EER32197.1 | |
| *A. nidulans* | 3 | (20.00) | 0 | (0.00) | | EAA58119.1 | |
| *Y. lipolytica* | 3 | (21.43) | 0 | (0.00) | CAG79453.1 | CAG79234.1 | |
| *N. crassa* | 2 | (13.33) | 0 | (0.00) | ESA42867.1 | ESA42646.1 | |
| *S. japonicus* | 1 | (6.67) | 0 | (0.00) | EEB05964.1 | EEB09703.2 | |
| *S. octosporus* | 1 | (6.67) | 0 | (0.00) | EPX4426.1 | EPX71342.1 | |
| *S. pombe* | 1 | (6.67) | 0 | (0.00) | CAA91078.1 | CAB52735.1 | |

## Fluorescence microscopy

Cultures were grown in 25 ml of SD medium (or 25 ml of SD medium missing uracil in experiments with plasmids) in 125 ml conical flasks shaking at 200 rpm and 30°C until they reached mid-log phase (OD600 between 0.3 and 0.4). The cultures were then diluted to $OD_{600}$ 0.1 in 25 ml of fresh medium and allowed to grow to an OD600 of 0.4. At this point, 300 µl samples were transferred to a chamber slide (Nunc Lab-Tek II #155409) that had been treated with 2 mg/ml concanavilin A. The slides were then washed three times with 300 µl treatment medium (either SD, SD + KCl, or SD-Glu, -N, or -Glu + N) and loaded into a 30o°C chamber.

Images were acquired using a Deltavision Elite Microscope equipped with an Olympus 100x, 1.4NA, objective and a sCMOS camera. We collected a Z-series of 5 images with 1 µm spacing in the YFP (YFP filter; Ex. 496–528 nm, Em. 537–559 nm), GFP (GFP filter; Ex. 425–––495 nm, Em. 500–550 nm), and RFP (mCherry filter; Ex. 555–590 nm, Em. 600-675nm) channels at each time-point to ensure that all of the fluorescent bodies in the cell were detected. Image files were then processed

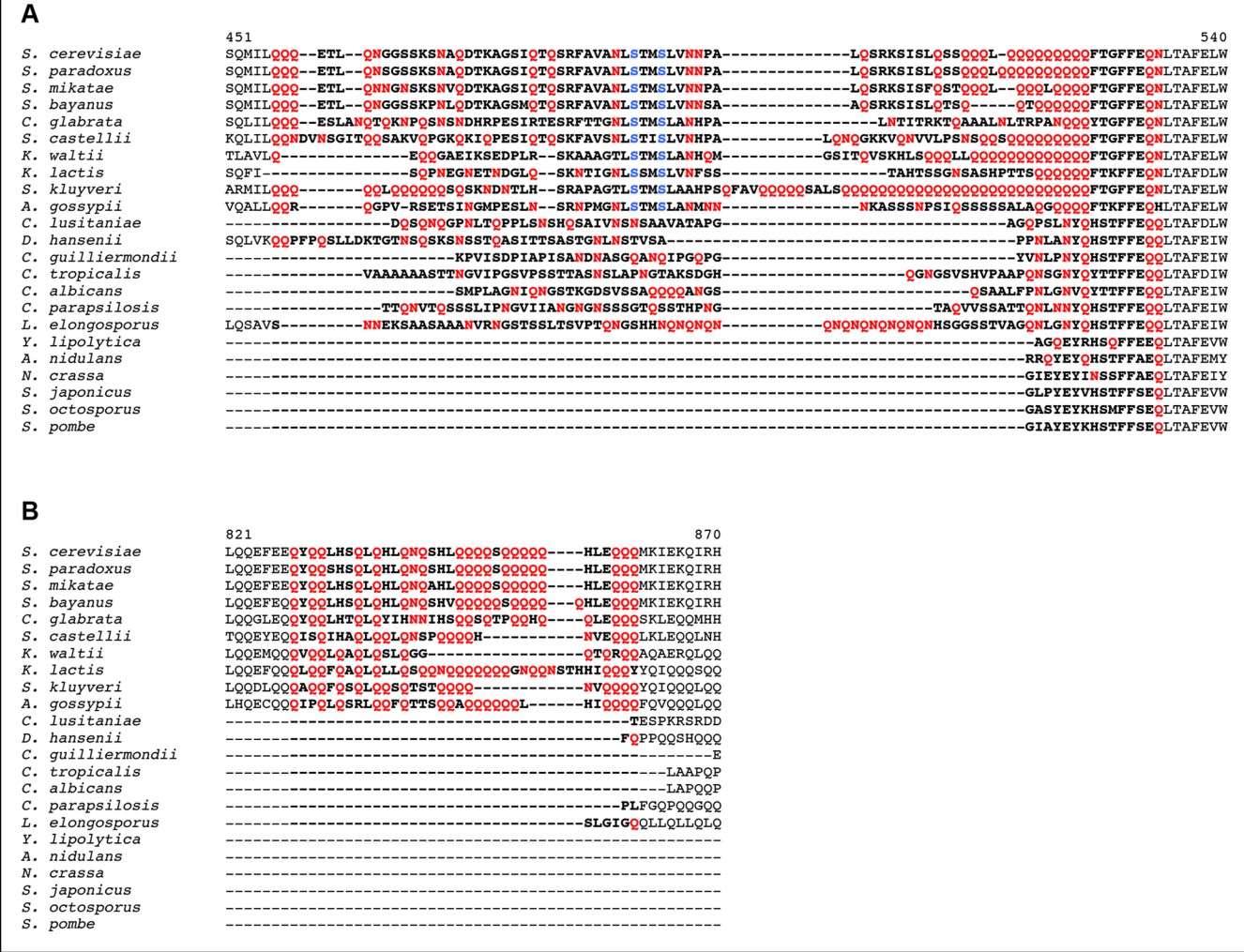

**Figure 6.** Partial alignment of Kog1 sequences from S. *cerevisiae* and 22 fungal orthologs. (**A**) PriLM1 and (**B**) PriLM2 (shown in bold) were defined using a hidden Markov Model trained on known yeast prion domains (*Alberti et al., 2009*). The Glutamine (Q) and Asparagine (N) residues in each PriLM are highlighted in red, while the Snf1 targets sites (Ser 491 and 494) are highlighted in blue. PriLMs 1 and 2 are both insertions between highly conserved domains in Kog1.

The following figure supplements are available for Figure 6:

**Figure supplement 1.** Partial alignment of Kog1 from S. *cerevisiae* with orthologous genes in higher eukaryotes.

in ImageJ to create the maximum projection from the stacks. Glucose repletion experiments (*Figure 1D*) were performed in a similar manner, but cells were collected by filtration and then transferred to medium without glucose for one hour before cells were loaded into the chamber slides.

We calculated the fraction of cells containing one or more Kog1 body by manually inspecting the images for >200 cells per time-point, per experiment. The time-course data was fit to a single exponential equation; $A*(1-e^{-t/\tau}) + c$ for starvation and $(A*(e^{-t/\tau}) + c$ for repletion experiments, where A is the fraction of cells that form bodies during the time-course, $\tau$ is the apparent time-constant, and c is fraction of cells that have bodies at the start (starvation) or end (repletion) of the time-course. In cases where there was no change in Kog1-body levels during the time-course (e.g. in SD medium) the data was fit to a line. All fitting was done in Igor Pro 6.3 (WaveMetrics) and the errors reported are the estimated standard deviation from the fit.

**Table 2.** *S. cerevisiae* strains used in this study.

| Strain | Catalogue Number | Description | Source |
|---|---|---|---|
| wt | ACY044 | W303 *MATa trp1-1 can1-100 leu2-3,112 his3-11,15 ura3 GAL +* | *Capaldi et al. 2008* |
| Kog1-YFP Vph1-mCh | ACY731 | [ACY044] *KOG1-YFP (HIS3) VPH1-mCherry (NAT +) MATa* | This Study |
| Tor1-GFP Vph1-mCh | ACY939 | [ACY044] *TOR1-D330 3xGFP VPH1-mCherry (NAT +) MATa* | This Study |
| Kog1-YFP | ACY173 | [ACY044] *KOG1-YFP (HIS3) MATa* | This Study |
| Kog1-YFP Pbp1-mCh | ACY959 | [ACY044] *KOG1-YFP (HIS3) pPBP1mCH (URA3) MATa* | This Study |
| Kog1-YFP Edc3-mCh | ACY856 | [ACY044] *KOG1-YFP (HIS3) pEDC3mCH (URA3) MATa* | This Study |
| *snf1Δ* Kog1-YFP | ACY743 | [ACY044] *SNF1::URA3 KOG1-YFP (HIS3) MATa* | This Study |
| Kog1-3xFlag | ACY161 | [ACY044] *KOG1-3xFLAG (HIS3) MATa* | This Study |
| S491A S494A Kog1-YFP | ACY871 | [ACY044] *KOG1$^{S491A\ S494A}$-YFP (HIS3) MATa* | This Study |
| PrDm1 Kog1-YFP | ACY844 | [ACY044] *KOG1$^{Q513-525A}$-YFP (HIS3) MATa* | This Study |
| PrDm2 Kog1-YFP | ACY768 | [ACY044] *KOG1$^{Q846-855A}$-YFP (HIS3) MATa* | This Study |
| PrDm1 + 2 Kog1-YFP | ACY878 | [ACY044] *KOG1$^{Q513-525A\ Q846-855A}$-YFP (HIS3) MATa* | This Study |
| Sch9-3HA | ACY359 | [ACY044] *SCH9-3HA (TRP1) MATa* | *Hughes Hallett et al. 2014* |
| *snf1Δ* Sch9-3HA | ACY537 | [ACY044] *SNF1::URA3 SCH9-3HA (TRP1) MATa* | *Hughes Hallett et al. 2014* |
| S491A S494A Sch9-3HA | ACY869 | [ACY044] *KOG1$^{S491A\ S494A}$SCH9-3HA (TRP1) MATa* | This Study |
| PrDm1 Sch9-3HA | ACY766 | [ACY044] *KOG1$^{Q513-525A}$SCH9-3HA (TRP1) MATa* | This Study |
| PrDm2 Sch9-3HA | ACY840 | [ACY044] *KOG1$^{Q846-855A}$SCH9-3HA (TRP1) MATa* | This Study |
| PrDm1 + 2 Sch9-3HA | ACY765 | [ACY044] *KOG1$^{Q513-525A\ Q846-855A}$SCH9-3HA (TRP1) MATa* | This Study |
| Tco89-YFP | ACY171 | [ACY044] *Tco89-YFP (HIS3) MATa* | This Study |
| PrDM1 + 2 Tco89-YFP | ACY783 | [ACY044] *KOG1$^{Q513-525A\ Q846-855A}$ Tco89-YFP (HIS3) MATa* | This Study |

| Plasmid | Catalogue Number | Description | Source |
|---|---|---|---|
| pEDC3mCH | ACB252 | [CEN] EDC3-mCherry (URA3) | *Buchan et al. 2008* |
| pPBP1mCH | ACB282 | [CEN] PBP1-mCherry (URA3) | *Shah et al. 2014* |

## Purification of Kog1/TORC1 and peptide mapping

Yeast expressing Kog1-3xFlag were inoculated into 50 ml of SD medium (for glucose starvation experiments) or 50 ml YEPD (for osmotic stress experiments) and grown overnight in 250 ml conical flasks, shaking at 200 rpm and 30°C. These cultures were then used to inoculate 2.5L of fresh media to OD$_{600}$ 0.1 and grown (as above) in 4L flasks until they reached an OD600 of 0.6. At this point cultures were either harvested by filtration and frozen in liquid nitrogen (SD sample), or subjected to 0.4M KCl stress, or glucose starvation for 5 min, and harvested by filtration and frozen in liquid nitrogen.

To isolate TORC1, cells were washed off the filter with 10 ml chilled lysis buffer (50 mM HEPES pH 7.1, 50 mM NaF, 10% glycerol, 150 mM KCl, 0.5 mM EDTA, 0.5 mM EGTA, 0.5% Tween 20, and 2 mM PMSF) and pelleted by centrifugation at 4°C. The cell pellets were then resuspended in chilled lysis buffer supplemented with a protease inhibitor cocktail (Sigma Aldrich, St. Louis, MO; P8215), split into 100 OD unit aliquots, and lysed by bead beating (5 X 20 s with 30 s intervals on ice). The lysate was then collected after centrifugation for 5min at 3000rpm and transferred into fresh tubes before being spun down again at 500 g for 5min at 4°C. The clarified lysate was then incubated with 100 µl protein G Dynal beads + 10 µl anti-Flag antibody for 30min at 4°C. The resin was washed 3 times for 1 min with 1 ml lysis buffer at 4°C and the protein eluted by boiling the beads in 150 µl SDS loading buffer for 5min, and loaded onto a 10% acrylamide gel (BioRad 161-1155). The gels were then stained using colloidal blue and the bands corresponding to Tor1 and Kog1 excised, subjected to protease treatment, and analyzed using Mass Spectrometry.

## Sch9 bandshift experiments

Bandshift measurements were performed using a modified version of the protocol developed by Urban and Loewith (*Urban et al., 2007*). Cultures were grown in conical flasks shaking at 200rpm and 30°C until mid log phase (OD$_{600}$ between 0.55 and 0.6). At this point, a 47 ml sample was collected, mixed with 3 ml 100% Trichloroacetic acid (TCA), and held on ice for at least 30 min (and up to 6 hrs). The remaining culture was then collected by filtration, transferred to the appropriate medium after one wash with 100ml of -Glu medium, and further samples collected in TCA, as described above. The samples were then centrifuged at 4000rpm for 5 min at 4°C, washed twice with 4°C water, twice with acetone, and disrupted by sonication at 15% amplitude for 5s before centrifugation at 8000rpm for 30s. The cell pellets were then dried in a speedvac for 10 min at room temperature, and frozen until required at -20°C.

Protein extraction was performed by bead beating (6 x 1 min, full speed) in urea buffer (6M Urea, 50 mM Tris-HCl pH 7.5, 5 mM EDTA, 1 mM PMSF, 5 mM NaF, 5 mM NaN$_3$, 5 mM NaH$_2$PO$_4$, 5 mM p-nitrophenylphosphate, 5 mM β-glycerophosphate, 1% SDS) supplemented with complete protease and phosphatase inhibitor tablets (Roche, Indianapolis, IN; 04693159001 and 04906845001). The lysate was then collected after centrifugation for 5 min at 3000 rpm, resuspended into a homogenous slurry, and heated at 65°C for 10 min. Soluble proteins were then separated from insoluble cell debris by centrifugation at 15,000 rpm for 5 min, and the lysate stored at -80°C until required. Samples were then subjected to cleavage by 2-Nitro-5-thiocyanatobenzoic acid (NTCB) for 12–16 hr at room temperature (1 mM NTCB and 100 mM CHES, pH 10.5) before further analysis. Cell extracts were then heated to 95°C in SDS sample buffer for 5 min before they were run on an SDS-PAGE gel, transferred to a nitrocellulose membrane. Western blotting was carried out using 12CA5 (anti-HA) and an anti-mouse secondary labeled with a IRDye 800CW (LiCor) and the blots scanned using a LiCor Odyssey Scanner (LiCor, Lincon, NE).

Protein mobility shift data were quantified using a custom MATLAB script, described previously in (*Hughes Hallett et al., 2014*). Briefly, the data in each lane of the gel was simplified by calculating the average signal intensity at each position along the length of the lane, and normalized so that the total signal matches that of the t = 0 control. The normalized data were then used to calculate a position weighted mean for each lane. This was done by defining the first pixel/data point in the gel as 100% phosphorylated and the last pixel in the gel as 0% phosphorylated. Pixels in-between these points were weighted (between 100 and 0) based on their position relative to the top and bottom of the gel (on a linear scale). The values for each position (pixel) were then summed to calculate the total amount of protein phosphorylation in a sample. Finally, since these numbers are in arbitrary units, we normalized all of the values using those found in the wild-type strain under the same conditions. This was done by multiplying the mean phosphorylation values in each strain by constant A, and adding a constant B, so that the values in the wild-type strain at time = 0 min (SD, 2% glucose) are 1.0 and time = 60 min (-G, 0% glucose) are 0.0. Error bars were then scaled using the same constants.

## Mass spectrometry

Gel slices were washed for 15 min each with water, 50/50 acetonitrile/water, acetonitrile, 100 mM ammonium bicarbonate, followed by 50/50 acetonitrile/100 mM ammonium bicarbonate. The solution was then removed and the gel slices dried by vacuum centrifugation. Next, the dried gel slices were reduced by covering them with 10 mM dithiothreitol in 100 mM ammonium bicarbonate and heating them at 56°C for 45 min; alkylated by covering them with a solution of 55 mM iodoacetamide in 100 mM ammonium bicarbonate and incubating in the dark at ambient temperature for 30min, and washed with 100 mM ammonium bicarbonate for 10 min and 50 mM ammonium bicarbonate + 50% acetonitrile for 10 min. The gel slices were then dried again and treated with an ice-cold solution of 12.5 ng/μL trypsin (Promega, Madison, WI) in 100 mM ammonium bicarbonate. After 45 min, the trypsin solution was removed, discarded, and a volume of 50 mM ammonium bicarbonate was added to cover the gel slices and they were incubated overnight at 37°C with mixing on a shaker. Samples were then spun down in a microfuge and the supernatant collected. The same Gel slices were then incubated in 0.1% trifluoroacetic acid (TFA) and acetonitrile, centrifuged, and the supernatant collected. At this point, the digestion supernatant and the extraction supernatant were pooled, split into two tubes, and concentrated using vacuum centrifugation. One tube was further

digested with thermolysin (Promega, Madison, WI) by resuspending the tryptically digested peptides with a solution containing 50 mM Tris-HCl pH 8 and 0.5 mM calcium chloride and adding 1 µg of thermolysin. Digestion was carried out at 75˚C with mixing for 5 hr. The thermolysin was quenched by adding TFA to a 0.5% final concentration. All samples were desalted using ZipTip $C_{18}$ (Millipore, Billerica, MA) and eluted with 70% acetonitrile/0.1% TFA. The desalted material was concentrated to dryness in a speed vac.

The proteolytically-digested samples were brought up in 20 µL of 2% acetonitrile in 0.1% formic acid and 18 µL and then analyzed by LC/ESI MS/MS with a Thermo Scientific Easy-nLC II (Thermo Scientific, Waltham, MA) coupled to a Orbitrap Elite ETD (Thermo Scientific, Waltham, MA) mass spectrometer using a trap-column configuration as described in (Licklider et al., 2002). In-line de-salting was accomplished using a reversed-phase trap column (100 µm × 20 mm) packed with Magic $C_{18}$AQ (5-µm, 200Å resin; Michrom Bioresources, Auburn, CA) followed by peptide separations on a reversed-phase column (75 µm × 250 mm) packed with Magic $C_{18}$AQ (5-µm, 100Å resin; Michrom Bioresources, Auburn, CA) directly mounted on the electrospray ion source. A 45-min gradient from 2% to 35% acetonitrile in 0.1% formic acid at a flow rate of 400 nL/min was used for chromatographic separations. A spray voltage of 2500V was applied to the electrospray tip and the Orbitrap Elite instrument was operated in the data-dependent mode, switching automatically between MS survey scans in the Orbitrap (AGC target value 1,000,000, resolution 120,000, and injection time 250 milliseconds) with collision induced dissociation MS/MS spectra acquisition in the linear ion trap (AGC target value of 10,000 and injection time 100 milliseconds), higher-energy collision induced dissociation (HCD) MS/MS spectra acquisition in the Orbitrap (AGC target value of 50,000, 15,000 resolution and injection time 250 milliseconds) and electron transfer dissociation (ETD) MS/MS spectra acquisition in the Orbitrap (AGC target value of 50,000, 15,000 resolution and injection time 250 milliseconds). The three most intense precursor ions from the Fourier-transform (FT) full scan were consecutively selected for fragmentation in the linear ion trap by CID with a normalized collision energy of 35%, fragmentation in the HCD cell with normalized collision energy of 35%, and fragmentation by ETD with 100 ms activation time. Selected ions were dynamically excluded for 30 seconds.

Data analysis was performed using Proteome Discoverer 1.4 (Thermo Scientific, San Jose, CA). The data were searched against the Saccharomyces Genome Database (downloaded 02/03/2011; www.yeastgenome.org) that was appended with protein sequences from the common Repository of Adventitious Proteins or cRAP (www.thegpm.org/crap/). Two searches were performed corresponding to the proteolytic enzymes trypsin, and trypsin/thermolysin (no enzyme selected). The Maximum missed cleavages was set to 2. The precursor ion tolerance was set to 10 ppm and the fragment ion tolerance was set to 0.8 Da. Variable modifications included oxidation on methionine (+ 15.995 Da), carbamidomethyl (+ 57.021 Da) on cysteine, and phosphorylation on serine, threonine, and tyrosine (+ 79.996 Da). Sequest HT was used for database searching. PhosphoRS 3.1 (Taus et al., 2011) was used for assigning phosphosite localization probabilities. All search results were run through PSM Validator for false discovery rate evaluation of the identified peptides.

## Acknowledgments

We thank Claudio DeVirgilio for sharing the Tor-3xGFP strain; Roy Parker for sharing the Edc3-mCherry plasmid; Paul Herman for sharing the Pbp1-mCherry plasmid; and Ross Buchan, Tricia Serio, and members of the Capaldi laboratory for critical reading of the manuscript. We also thank Phil Gafken and Lisa Jones of the Fred Hutchinson Cancer Research Center's Proteomics Facility for carrying out the phosphopeptide mapping experiments. This work was supported by the National Institutes of Health (NIH) grants 1R01GM097329 and 5T32GM008659.

## Additional information

### Funding

| Funder | Grant reference number | Author |
|---|---|---|
| National Institutes of Health | 1R01GM097329 | Andrew P Capaldi |
| National Institutes of Health | 5T32GM008659 | James E Hughes Hallett |

The funders had no role in study design, data collection and interpretation, or the decision to submit the work for publication.

## Author contributions

JEHH, Conception and design, Acquisition of data, Analysis and interpretation of data, Drafting or revising the article, Contributed unpublished essential data or reagents; XL, Acquisition of data, Contributed unpublished essential data or reagents; APC, Conception and design, Analysis and interpretation of data, Drafting or revising the article

## Additional files

### Supplementary files

• Source code 1. Bandshift Analysis.txt includes the MATLAB script used to quantify the bandshift data in *Figure 5*.

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
