## [Decision Letter]

Thank you for submitting your work entitled "AMPK/Snf1 Promotes the Formation of Kog1-bodies to Increase the Activation Threshold of TORC1 in Budding Yeast" for peer review at *eLife*. Your submission has been favorably evaluated by Tony Hunter (Senior Editor) and three reviewers, one of whom is a member of our Board of Reviewing Editors.

The reviewers have discussed the reviews with one another and the Reviewing Editor has drafted this decision to help you prepare a revised submission.

All the reviewers found the work interesting and, in principle, suitable for *eLife*. However, they also noted some weaknesses, particularly the biological role of the Kog1 bodies, whether Sch9 phosphorylation status is truly a measure of TORC activity, and whether Kog1 bodies really affect TORC activity. The Reviewers made individual points and suggestions, but the overall comments are quite consistent. The most important request for the revised manuscript is to clearly connect the role of Kog1 bodies with TORC function. Instead of listing specific experiments that are absolutely required vs. others that would be useful, we request the authors address the points raised in the reviews, and do whatever experiments they judge best that will answer some of these issues (and are feasible in a reasonable timeframe). We don't expect or require the authors to address all the issues with new experiments, but we do ask that some of the "holes" be filled. We stress that the purpose of these new experiments is not simply to add more material, but rather to address specific issues raised.

*Reviewer #1:* Although I'm not an expert in this area, this appears to be an interesting, well-done, and well written paper that, in principle, should be published in *eLife*. The authors convincingly describe a new mechanism in which glucose or nitrogen starvation results in disassembly of TORC in which the Kog/raptor subunit forms Kog1 bodies near the edge of the vacuole. This relocalization requires Snf1/AMP kinase-mediated phosphorylation of Kog1 at Ser491/494 and two nearby prion-like motifs. Finally, the authors provide evidence that the Kog1-bodies increase the threshold for TORC activation to help cells remain in the starved state unless nutrients are fully restored. This is a new mechanism of clear physiological importance in yeast. While it is unknown if this mechanism is utilized elsewhere, this should not affect its suitability for *eLife*. It is highly unlikely that this or a related mechanism is specific to *S. cereivisae*, particularly the general concept of why protein bodies might be functionally important. There is one weakness of the paper that I would like to see addressed and another experiment that would be nice.

1) The experiment in Figure 5 showing increased Sch9 phosphorylation in mutant strains is critical for the conclusion about the biological role of Kog1 bodies. The 2-fold effect is clear, but small. The authors argue that the effect is really much bigger because Kog1-bodies only occur in about half of the wild-type cells. This is a reasonable argument, but it is just an argument. It would be very nice to address this in a single-cell manner by simultaneously assaying Kog1-body formation and Sch9 phorylation (appropriate phospho-antibody probably or some other activity assay). In this way, the relationship between Kog1-body formation and reduced Sch9 phosphorylation would be more clearly demonstrated.

2) Of less importance, but also useful, it would be nice to examine a Ser491/494 derivative with both serines converted to either glutamate or aspartate. It is quite possible that this would lead to constitutive Kog1-body formation and hence enhance the paper. In addition, it would be interesting to know what fraction of Kog1 is incorporated into the Kog1 bodies, and whether other components of TORC1 are sequestered in these structures.

*Reviewer #2:* This study examines the intracellular fate of two key components of yeast TORC1, namely, Tor1 and Kog1, following nutrient starvation using fluorescence microscopy. The main finding is that glucose starvation triggers a change in localization of Kog1 from a peri-vacuolar location to a punctate vacuole-associated location termed "Kog1 bodies". This change in Kog1 localization is determined to correlate to varying degrees with Kog1-dependent phosphorylation by Snf1, the presumptive ortholog of AMPK in yeast, and decreased TORC1 activity during glucose starvation, as monitored by Sch9 phosphorylation.

The strengths of the manuscript include the novel finding of a change in localization of an essential TORC1 component in a nutrient responsive manner and the fact that all of the data reported are solid and well controlled. There are two major concerns that need to be addressed before publication can be considered:

1) The physiological significance of these findings is unclear. Thus, the authors suggest that TORC1 activity is dampened by glucose starvation via Kog1 body formation, but the data presented do not address this directly, and some of the data using Sch9 as a readout are inconclusive, in particular with the Snf1 phosphorylation site mutants. It will be important to demonstrate that cells expressing the Kog1 mutants have reduced fitness following either prolonged glucose starvation and/or return to glucose replete conditions. One possibility would be to test phosphomimetic substitutions to see if an inability to dephosphorylate Kog1 leads to decreased viability in low glucose. In general, because the mutants examined do not have any obvious phenotype with respect to glucose starvation per se, enthusiasm for these findings is somewhat dampened.

2) The relationship between Snf1 and Kog1 phosphorylation needs to be defined. The authors appear to imply that Kog1 is a direct target for Snf1 yet state in the Discussion this is likely to be mediated by another kinase. It should be noted that the yeast literature has remained quite confusing and controversial in terms of the role of Snf1/TORC1 in comparison to AMPK and mTORC1. As it stands, the present work does not clarify the yeast relationship, and it's important not to muddy the waters any further. If Snf1 is not the kinase that phosphorylates Kog1, do the authors have an idea as to the relevant kinase involved?

*Reviewer #3:* This manuscript presents an interesting story that, if correct, seems like an important advance on an important and significant issue: the mechanisms and consequences of regulation of the nutrient-sensing protein kinase TORC1. The authors describe experiments that they claim show that:

1) the TORC1 component Kog1 rapidly (in ~ 10 min) aggregates (reversibly) into "bodies" (that are not stress granules or P-bodies, (Figure 2)) when starved for glucose (Figure 1);

2) formation of Kog1 bodies depends on Snf1 and on two Snf1-dependent phosphorylation sites in Kog1 (Figure 3), and on;

3) two prion-like domains in Kog1 (Figure 4);

4) Kog1-body formation decreases TORC1 activity and ensures that cells remain in a starved state until sufficient glucose becomes available (Figure 5).

Conclusions #1-3 seem solid and supported by unimpeachable evidence. The nub of the story is conclusion #4, supported by evidence presented in the section "Kog-1 body formation increases the threshold for TORC1 activation" (Figure 5). This is where the story gets confusing. My main problem is that it is not clear to me that the rapid dephosphorylation of Sch9 (a TORC1 target) is measuring TORC1 activity. The rapidity of this event suggests to me that it could be due to an event other than inactivation of TORC1 (activation of a protein phosphatase?). This is key, because the authors' main conclusion is that Kog1-body formation regulates TORC1 function, and Sch9 dephosphorylation is their assay for TORC1 function. Is that really reporting TORC1 function? Again, the "discovery" "…that TORC1 inhibition precedes Kog1-body formation…" depends on Sch9 dephosphorylation being an indicator of TORC1 function. Is it? Also, "Instead, it appears that Snf1 regulates TORC1 both by phosphorylating Kog1 at Ser491 and 494, and through an additional, unknown mechanism" is opaque to me.

But what confuses me is the observation that addition to cells of a large amount of glucose (2%) rapidly activates TORC1 (indicated by phosphorylation of Sch9 (Figure 5)). Doesn't this require Kog1 to move out of the "bodies" and back into TORC1? But that is a relatively slow process (Figure 1). The authors' suggest that addition to cells of small amounts of glucose (0.02% - 0.1%) results in even slower movement of Kog1 from "bodies" to TORC1, and that's true for 0.02%, but not for 0.1% (Figure 1). I think these discrepancies between the rate of disassembly of Kog1-bodies and the rate of TORC1 activation (if that's really what Sch9 phosphorylation is measuring) are the main weakness of the paper.

In summary, the authors state their conclusions in the first sentence of the Discussion [my critique inserted]: "In this study we show that glucose starvation leads to inactivation of TORC1 signaling [only if Sch9 dephosphorylation is actually reporting TORC1 inactivation], disassociation of the Kog1-Tor complex [I don't think I saw any evidence of that in the manuscript], and the formation of Kog1-bodies [yes, that's clear; the evidence for that is unimpeachable]."

(The suggestion that the prion-like domains of Kog1 and the tuberous sclerosis complex have analogous roles is speculative but interesting, and quite appropriate for the Discussion. I also found the speculation in the last paragraph of the Discussion about the "role that protein body formation plays in a cell" interesting and potentially significant.)

---

## [Author Response]

*All the reviewers found the work interesting and, in principle, suitable for* eLife*. However, they also noted some weaknesses, particularly the biological role of the Kog1 bodies, whether Sch9 phosphorylation status is truly a measure of TORC activity, and whether Kog1 bodies really affect TORC activity. The Reviewers made individual points and suggestions, but the overall comments are quite consistent. The most important request for the revised manuscript is to clearly connect the role of Kog1 bodies with TORC function. Instead of listing specific experiments that are absolutely required vs. others that would be useful, we request the authors address the points raised in the reviews, and do whatever experiments they judge best that will answer some of these issues (and are feasible in a reasonable timeframe). We don't expect or require the authors to address all the issues with new experiments, but we do ask that some of the "holes" be filled. We stress that the purpose of these new experiments is not simply to add more material, but rather to address specific issues raised.* Reviewer #1:

*Although I'm not an expert in this area, this appears to be an interesting, well-done, and well written paper that, in principle, should be published in* eLife*. The authors convincingly describe a new mechanism in which glucose or nitrogen starvation results in disassembly of TORC in which the Kog/raptor subunit forms Kog1 bodies near the edge of the vacuole. This relocalization requires Snf1/AMP kinase-mediated phosphorylation of Kog1 at Ser491/494 and two nearby prion-like motifs. Finally, the authors provide evidence that the Kog1-bodies increase the threshold for TORC activation to help cells remain in the starved state unless nutrients are fully restored. This is a new mechanism of clear physiological importance in yeast. While it is unknown if this mechanism is utilized elsewhere, this should not affect its suitability for* eLife*. It is highly unlikely that this or a related mechanism is specific to* S. cereivisae*, particularly the general concept of why protein bodies might be functionally important. There is one weakness of the paper that I would like to see addressed and another experiment that would be nice. 1) The experiment in Figure 5 showing increased Sch9 phosphorylation in mutant strains is critical for the conclusion about the biological role of Kog1 bodies. The 2-fold effect is clear, but small. The authors argue that the effect is really much bigger because Kog1-bodies only occur in about half of the wild-type cells. This is a reasonable argument, but it is just an argument. It would be very nice to address this in a single-cell manner by simultaneously assaying Kog1-body formation and Sch9 phorylation (appropriate phospho-antibody probably or some other activity assay). In this way, the relationship between Kog1-body formation and reduced Sch9 phosphorylation would be more clearly demonstrated.* This is a good suggestion and we have been trying to perform experiments like this, but have not been able to measure TORC1 activity in a single cell (and to the best of our knowledge nor has anyone else working in yeast). However, we would argue that the two-fold effect is already a quite large effect given the importance of TORC1 activity for cell growth, and is enough to establish a function of the Kog1-bodies – something that has yet to be done for other bodies/granules as we mention in the Discussion.

*2) Of less importance, but also useful, it would be nice to examine a Ser491/494 derivative with both serines converted to either glutamate or aspartate. It is quite possible that this would lead to constitutive Kog1-body formation and hence enhance the paper.*

As described above, we have now made the DD and EE mutants and find that they are not viable – indicating that Kog1-bodies are incompatible with growth. These data are incorporated into the text and supplement.

*In addition, it would be interesting to know what fraction of Kog1 is incorporated into the Kog1 bodies,*

We agree that it would be interesting to know what fraction of Kog1 molecules go to bodies, but unfortunately we have not been able to calculate this number. Kog1 levels are so low (a few hundred molecules per cell) that we cannot accurately measure the amount of Kog1 distributed around the vacuole and/or in the cytoplasm, either before or after glucose starvation – the Kog1 fluorescence levels are simply too close to the level of autofluorescence (Figure 1).

*and whether other components of TORC1 are sequestered in these structures.*

We have now examined localization of the other unique component of TORC1, Tco89, in glucose starvation conditions. This protein does move to bodies after 60 min, and this movement is limited in a strain where Kog1 bodies are slow to form (PriDm1+2). This is now mentioned in the text (first paragraph of the Results) and the data shown in the supplement (Figure 1—figure supplement 1).

Reviewer #2:

*This study examines the intracellular fate of two key components of yeast TORC1, namely, Tor1 and Kog1, following nutrient starvation using fluorescence microscopy. The main finding is that glucose starvation triggers a change in localization of Kog1 from a peri-vacuolar location to a punctate vacuole-associated location termed "Kog1 bodies". This change in Kog1 localization is determined to correlate to varying degrees with Kog1-dependent phosphorylation by Snf1, the presumptive ortholog of AMPK in yeast, and decreased TORC1 activity during glucose starvation, as monitored by Sch9 phosphorylation. The strengths of the manuscript include the novel finding of a change in localization of an essential TORC1 component in a nutrient responsive manner and the fact that all of the data reported are solid and well controlled. There are two major concerns that need to be addressed before publication can be considered: 1) The physiological significance of these findings is unclear. Thus, the authors suggest that TORC1 activity is dampened by glucose starvation via Kog1 body formation, but the data presented do not address this directly, and some of the data using Sch9 as a readout are inconclusive, in particular with the Snf1 phosphorylation site mutants. It will be important to demonstrate that cells expressing the Kog1 mutants have reduced fitness following either prolonged glucose starvation and/or return to glucose replete conditions. One possibility would be to test phosphomimetic substitutions to see if an inability to dephosphorylate Kog1 leads to decreased viability in low glucose. In general, because the mutants examined do not have any obvious phenotype with respect to glucose starvation per se, enthusiasm for these findings is somewhat dampened.*

We have now tested phosphomimetic substitutions, and as described above, find that strains with Ser 491 and 494 mutated to aspartic acid or glutamic acid do not survive/grow.

*2) The relationship between Snf1 and Kog1 phosphorylation needs to be defined. The authors appear to imply that Kog1 is a direct target for Snf1 yet state in the Discussion this is likely to be mediated by another kinase.*

We did not mean to imply that Snf1 directly phosphorylates Kog1, and to make sure there was no room for confusion, directly state that we believe Snf1 acts through another kinase to phosphorylate Kog1 in the Discussion. To make sure this is clear in other parts of the paper we have now adjusted the language in several places where we refer to Snf1 activating phosphorylation of Kog1.

For example in the last paragraph of the Introduction: Following up on these observations we show that the AMPK, Snf1, drives Kog1 into bodies by phosphorylating Kog1 on Ser 491 and 494.

Has been changed to: Following up on these observations we show that the AMPK, Snf1, drives Kog1 into bodies by phosphorylating, or triggering the phosphorylation of, Kog1 at Ser 491 and 494.

The other parts of the text where we discuss Snf1 and Kog1 phosphorylation already use appropriate wording, such as in this excerpt from Results:.

…These data showed that Kog1 is phosphorylated on Ser 491 and 494 in glucose starvation conditions, but not in osmotic stress or SD medium (Figure 3—figure supplement 1 and Figure 3—figure supplement 2). A short time later, Young and coworkers showed that Ser 491 and 494 on Kog1 are Snf1-dependent phosphorylation sites (Braun et al., 2014). Therefore, to test if Snf1 drives Kog1-body formation by promoting phosphorylation of Ser 491 and 494, we constructed a strain carrying Kog1^S491A/S494A^-YFP at the native locus and monitored Kog1 localization in glucose starvation conditions. We found that Kog1^S491A/S494A^ forms bodies slowly (τ=309 ± 48 min; Figure. 3B), and at a rate similar to that found in the *snf1*△ strain (τ=219 ± 21 min; Figure 3). Thus, Snf1 increases the rate of Kog1 body formation by phosphorylating, or triggering phosphorylation, of Kog1 in glucose starvation conditions.

*It should be noted that the yeast literature has remained quite confusing and controversial in terms of the role of Snf1/TORC1 in comparison to AMPK and mTORC1. As it stands, the present work does not clarify the yeast relationship, and it's important not to muddy the waters any further. If Snf1 is not the kinase that phosphorylates Kog1, do the authors have an idea as to the relevant kinase involved?*

We disagree with the statement that our work does not clarify the relationship between Snf1/AMPK and TORC1. The Loewith lab has shown that the AMPK target sites in Raptor do not play an obvious role in yeast. Here, we show that Snf1 activation causes phosphorylation of Ser491/494 in yeast – sites that don't exist in flies, mice or humans. Furthermore, we have shown that these sites have a unique function, driving Kog1 body formation. So we have shown that while Snf1/AMPK regulates TORC1 in humans and yeast, the mechanisms are different. This is an important advance. As the reviewer points out there is still more to learn; we need to identify the direct regulator(s) of Ser 491 and 494, and this is something we are working on. But at this point we do not know which kinase(s) act downstream of Snf1 to regulate.

Reviewer #3:

*This manuscript presents an interesting story that, if correct, seems like an important advance on an important and significant issue: the mechanisms and consequences of regulation of the nutrient-sensing protein kinase TORC1. The authors describe experiments that they claim show that: 1) the TORC1 component Kog1 rapidly (in ~ 10 min) aggregates (reversibly) into "bodies" (that are not stress granules or P-bodies, (Figure 2)) when starved for glucose (Figure 1); 2) formation of Kog1 bodies depends on Snf1 and on two Snf1-dependent phosphorylation sites in Kog1 (Figure 3), and on; 3) two prion-like domains in Kog1 (Figure 4); 4) Kog1-body formation decreases TORC1 activity and ensures that cells remain in a starved state until sufficient glucose becomes available (Figure 5). Conclusions #1-3 seem solid and supported by unimpeachable evidence. The nub of the story is conclusion #4, supported by evidence presented in the section "Kog-1 body formation increases the threshold for TORC1 activation" (Figure 5). This is where the story gets confusing. My main problem is that it is not clear to me that the rapid dephosphorylation of Sch9 (a TORC1 target) is measuring TORC1 activity. The rapidity of this event suggests to me that it could be due to an event other than inactivation of TORC1 (activation of a protein phosphatase?).*

This is a good point, and as described above, we were too loose in our language. All we can conclude is that the rapid dephosphorylation of Sch9 shows that Kog1 body formation does not play a role in the initial inactivation of TORC1 pathway signaling, not TORC1 itself. Therefore, we have changed our language to refer to rapid inactivation of the TORC1 pathway, or rapid inactivation of TORC1-Sch9 signaling, not TORC1.

*This is key, because the authors' main conclusion is that Kog1-body formation regulates TORC1 function, and Sch9 dephosphorylation is their assay for TORC1 function. Is that really reporting TORC1 function?*

Our key conclusion (that Kog1 bodies increase the threshold for TORC1 activation) depends on experiments we did comparing the level of Sch9 phosphorylation in strains carrying mutations in TORC1 to the level of Sch9 phosphorylation in the wild-type strain. So in this part of the paper, where we show that mutations in TORC1 (Kog1) cause hyperphosphorylation of Sch9 in low glucose, the only reasonable explanation is that TORC1 itself is more active when Kog1 does not form bodies.

*Again, the "discovery" "…that TORC1 inhibition precedes Kog1-body formation…" depends on Sch9 dephosphorylation being an indicator of TORC1 function. Is it?*

As described above, this now reads TORC1 pathway inhibition precedes Kog1-body formation.

*Also, "Instead, it appears that Snf1 regulates TORC1 both by phosphorylating Kog1 at Ser491 and 494, and through an additional, unknown mechanism" is opaque to me.*

In this part of the text, we are referring to the fact that deletion of Snf1 partially inhibits the rapid dephosphorylation of Sch9 seen in glucose starvation conditions but removing the Snf1 dependent target sites in Kog1 does not. Thus, Snf1 must both promote rapid dephosphorylation of Sch9 through an unknown mechanism, and the formation of Kog1 bodies by triggering phosphorylation of Kog1 at Ser 491/494. This sentence has now been changed to… Instead, it appears that Snf1 regulates the TORC1 pathway by triggering Kog1 phosphorylation at Ser 491 and 494, and through an additional, unknown, mechanism.

*But what confuses me is the observation that addition to cells of a large amount of glucose (2%) rapidly activates TORC1 (indicated by phosphorylation of Sch9 (Figure 5)). Doesn't this require Kog1 to move out of the "bodies" and back into TORC1? But that is a relatively slow process (Figure 1). The authors' suggest that addition to cells of small amounts of glucose (0.02% - 0.1%) results in even slower movement of Kog1 from "bodies" to TORC1, and that's true for 0.02%, but not for 0.1% (Figure 1). I think these discrepancies between the rate of disassembly of Kog1-bodies and the rate of TORC1 activation (if that's really what Sch9 phosphorylation is measuring) are the main weakness of the paper.*

TORC1 is rapidly activated by saturating amounts of glucose in cells containing Kog1-bodies, but this does not suggest, or require, that Kog1 moves back into TORC1 on a rapid time-scale. In fact, as Reviewer 3 points out, we know that this does not happen – Kog1 bodies disassemble on the hour timescale. Instead, we favor the following model:

Only some (let’s say 80-90%) of the Kog1 in a cell goes into bodies during glucose starvation (Figure 1). Thus, in the wild-type strain, there are a small number of functional, intact, TORC1 molecules in glucose starvation conditions. When you add 0.02% or 0.1% glucose to starving cells, you get low-level activation of the small number of intact TORC1 molecules, but it is not enough to trigger significant phosphorylation of Sch9 since you are fighting against a phosphatase. In contrast, when you add saturating amounts of glucose to starving cells you get high-level activation of the small number of TORC1 molecules, and that is enough to drive rapid phosphorylation of Sch9.

In the PriDm1, PriDm1+2, Snf1 delete, and S491/494 mutants, we have increased the number of intact TORC1 molecules by blocking Kog1-body formation, and as a result 0.02 and 0.1% glucose signals lead to low-level activation of a larger pool of TORC1 molecules – enough to activate Sch9.

We already refer to this idea in the Discussion when we say:

In this study we show that glucose starvation leads to rapid inactivation of TORC1-Sch9 signaling (τ<2.5 min, Figure 5), dissociation of the Kog1-Tor1 complex, and the formation of Kog1-bodies (τ=11 ± 3 min). Kog1-bodies then act to increase the threshold for TORC1 activation – probably by limiting the number of intact TORC1 molecules in the cell.

*In summary, the authors state their conclusions in the first sentence of the Discussion [my critique inserted]: "In this study we show that glucose starvation leads to inactivation of TORC1 signaling [only if Sch9 dephosphorylation is actually reporting TORC1 inactivation], disassociation of the Kog1-Tor complex [I don't think I saw any evidence of that in the manuscript], and the formation of Kog1-bodies [yes, that's clear; the evidence for that is unimpeachable]."*

As discussed above, we agree with the first point the reviewer brings up; we cannot say that glucose starvation leads to inactivation of TORC1 signaling, and instead now say that glucose starvation leads to rapid inactivation of TORC1-Sch9 signaling (τ<2.5 min, Figure 5), dissociation of the Kog1-Tor1 complex, and the formation of Kog1-bodies (τ=11 ± 3 min).

We do not agree with the second critique. In Figure 1 we clearly show that Kog1 goes to bodies, but Tor1 does not – thus TORC1 dissociates in glucose starvation conditions.